# Deep-Learning-Based Detection of Paroxysmal Supraventricular Tachycardia Using Sinus-Rhythm Electrocardiograms

**DOI:** 10.3390/jcm11154578

**Published:** 2022-08-05

**Authors:** Lei Wang, Shipeng Dang, Shuangxiong Chen, Jin-Yu Sun, Ru-Xing Wang, Feng Pan

**Affiliations:** 1Key Laboratory of Advanced Process Control for Light Industry (Ministry of Education), Jiangnan University, Wuxi 214122, China; 2Department of Cardiology, The Affiliated Wuxi People’s Hospital of Nanjing Medical University, Wuxi 214023, China; 3Department of Cardiology, The Affiliated Hospital of Nanjing Medical University, Nanjing 210011, China

**Keywords:** deep learning, convolution neural network, paroxysmal supraventricular tachycardia, electrocardiography, detection

## Abstract

Background: Paroxysmal supraventricular tachycardia (PSVT) is a common arrhythmia associated with palpitation and a decline in quality of life. However, it is undetectable with sinus-rhythmic ECGs when patients are not in the symptomatic onset stage. Methods: In the current study, a convolution neural network (CNN) was trained with normal-sinus-rhythm standard 12-lead electrocardiographs (ECGs) of negative control patients and PSVT patients to identify patients with unrecognized PSVT. PSVT refers to atrioventricular nodal reentry tachycardia or atrioventricular reentry tachycardia based on a concealed accessory pathway as confirmed by electrophysiological procedure. Negative control group data were obtained from 5107 patients with at least one normal sinus-rhythmic ECG without any palpitation symptoms. All ECGs were randomly allocated to the training, validation and testing datasets in a 7:1:2 ratio. Model performance was evaluated on the testing dataset through F1 score, overall accuracy, area under the curve, sensitivity, specificity and precision. Results: We retrospectively enrolled 407 sinus-rhythm ECGs of PSVT procedural patients and 1794 ECGs of control patients. A total of 2201 ECGs were randomly divided into training (n = 1541), validation (n = 220) and testing (n = 440) datasets. In the testing dataset, the CNN algorithm showed an overall accuracy of 95.5%, sensitivity of 90.2%, specificity of 96.6% and precision of 86.0%. Conclusion: Our study reveals that a well-trained CNN algorithm may be a rapid, effective, inexpensive and reliable method to contribute to the detection of PSVT.

## 1. Introduction

Paroxysmal supraventricular tachycardia (PSVT) is one of the common types of arrhythmia, a clinical syndrome characterized by the presence of regular and rapid tachycardia with abrupt onset and termination and an incidence of 35 per 100,000 patient-years [1,2]. The most frequent symptoms of PSVT are rapid heart rate, palpitations, lightheadedness, shortness of breath, chest pain, anxiety, and potentially syncope [3,4]. The primary consequence of PSVT in most patients is a decline in quality of life. In some rare cases, incessant PSVT can cause tachycardia-induced cardiomyopathy and sudden death [5]. PSVT is also a novel risk-factor for cryptogenic stroke [3,6,7]. Most patients with PSVT can be temporarily managed using physiological maneuvers, medications and occasionally even electrical cardioversion [5]. Catheter ablation is a permanent method for patients who want to cure the arrhythmia [8].

In the present study, the term “PSVT” mainly refers to atrioventricular nodal reentry tachycardia (AVNRT) or atrioventricular reentry tachycardia (AVRT) excluding atrial fibrillation, atrial flutter and atrial tachycardia. The occurrence of AVRT is based on an overt (Wolff–Parkinson–White (WPW) syndrome) or concealed accessory pathway [8,9]. A confirmatory diagnosis of PSVT can be made via electrocardiogram (ECG) or other rhythm-recording devices during tachycardia. Except for WPW syndrome, confirming a diagnosis of PSVT can be challenging because of its variable duration and sporadic nature [4]. Furthermore, patients with unrecognized AVNRT or AVRT based on a concealed accessory pathway cannot be diagnosed when they are not in the PSVT onset stage via sinus-rhythmic ECGs. Currently, most medical image detection relies on the classical approach, which follows a three-step procedure (hand-crafted feature extraction, study and recognition) [10,11]. However, deep learning is an end-to-end deep neural network that can extract even subtle features automatically to achieve identification, classification and prediction [12]. It is unlike conventional machine learning, which needs manual feature extraction. In recent years, deep neural networks, especially convolutional neural networks (CNNs) have been shown to outperform classical machine learning approaches in most medical image analysis tasks. The application of a CNN to an ECG can predict cardiovascular diseases and even non-cardiovascular diseases such as serum potassium aberrations, anemia and sleep apnea [13,14,15,16,17]. CNN-enabled ECGs could identify patients with left ventricle dysfunction using a threshold of LVEF ≤ 35% or 50% according to transthoracic echocardiogram [18,19]. A previous study also reported that a well-trained CNN could detect patients with the electrocardiographic signature of atrial fibrillation present during normal sinus rhythm, which has practical implications for atrial fibrillation screening [20]. However, the application of CNN to screen patients for PSVT from normal sinus rhythmic ECGs has not been investigated.

In the current study, we hypothesized that a deep neural network could identify PSVT patients even if they were not in the PSVT onset stage. To test and verify our hypothesis, we trained, validated and tested a CNN model using patients with standard 12-lead normal sinus rhythmic ECGs.

## 2. Materials and Methods

### 2.1. Data Sources and Collection

Under the approval of the Ethics Committee of The Affiliated Wuxi People’s Hospital of Nanjing Medical University, 1714 ECGs of 1143 patients with PSVT were collected from 1 January 2013 to 31 August 2021, and 5365 ECGs of 5107 patients were also collected between 1 January, 2020 and 10 March, 2020 as a control group. All ECG images in both control and PSVT groups were digital, standard 10-s, 12-lead ECGs acquired at a sampling rate of 500 Hz using MAC 800 or 1200ST ECG machines (GE Healthcare). The bandwidth of filter setting was 0.16–40 Hz. All ECG images used in this study were overread and analyzed by two cardiologists, who corrected diagnostic labels as needed. This study was carried out in accordance with The Code of Ethics of the World Medical Association (Declaration of Helsinki). 

### 2.2. Identifying Study Groups and Processing ECG Data

We used ECG data to classify patients into two groups: patients positive for PSVT and patients without any symptoms, signs or records of PSVT. For the PSVT group, the inclusion criteria were as follows: (1) Patients were included if they had abrupt onset and termination of palpitation symptoms or they were diagnosed with PSVT clinically; (2) Patients were diagnosed and confirmed with PSVT by electrophysiological study and radiofrequency ablation; (3) Patient were included if they had a sinus rhythmic ECG before an electrophysiological procedure. For the control group, patients who were evaluated without evidence of PSVT in the outpatient clinic by cardiologist via history collection, medical records or telephone follow-up were included. The exclusion criteria for the control group were: patients with signs of PSVT and general exclusion criteria. The general exclusion criteria were as follows: non-sinus-rhythm ECGs, WPW syndrome, serious atrioventricular bundle block, wide QRS tachycardias, acute myocardial infarction, HR > 130 bpm, HR < 45 bpm and age < 14 years for both groups (Figure 1). The patients in both groups did not receive any anti-arrhythmic drugs. It is worth mentioning that a complete ECG not only contains a physiological signal waveform diagram but also some metadata such as sex, age, heart rate, P–R interval and QT interval, which will interfere with the feature extraction of the model. Therefore, the metadata part of the ECG images was cut out from the original ECG images (resolution of 6786 × 4731); only the physiological signal waveform diagram was kept, with a resolution of 6600 × 3347. 

In total, 2201 ECG images were randomly allocated to the training set, validation set and testing set at a ratio of 7:1:2. None of the patients overlapped among these three groups. ECGs in the training set were used for training the model, ECGs in the validation set were used to adjust parameters and optimize the model, and the remaining 20% of ECGs comprised the testing set, which was used to assess the generalization ability of the model and evaluate the ability of a CNN-based model to detect PSVT. A schematic representation of the proposed method is given in Figure 2.

### 2.3. The Proposed Deep Neural Network

The CNN was implemented by using PyTorch backend with Python, and all the experiments were conducted on a Windows Server 2012 R2 with an NVIDIA Tesla V100 (16 GB). Numpy, matplotlib and other deep learning libraries were used for deep learning algorithms.

All ECG images were resized to 1600 × 800 as inputs for the model through a quadratic linear interpolation scaling algorithm, with the aim of retaining as much waveform information as possible to help detect the subtle features. The batch size was set to 8. Adam optimizer and categorical cross entropy loss function were selected. For the hyper-parameters of the proposed model, the initial learning rate was set to 10^−4^. Meanwhile, cosine annealing was adopted for learning rate decay.

Considering the ECG images consisted of multiple waves, we paid more attention to edge detection, and we set three convolution layers in the initial stage of the network to extract waveform features, as the early layers of the neural network was aimed at detecting edges. Each convolution layer was followed by a batch-normalization layer, which was used to eliminate distribution differences between layers while preserving the sample distribution characteristics. Following the third batch-normalization layer, there was a nonlinear ReLU function and a max-pooling layer [21,22]. Then, the samples had better sparsity and the dimension of the array was reduced on the premise of preserving the characteristics of the sample. Moreover, the SE–ResNet bottleneck module was applied to extract subtle features that were not readily apparent to the naked eye from ECG images. The design of the bottleneck not only reduced the network parameters but also deepened the network depth to study more features. Meanwhile, an attention mechanism that allowed the network to emphasize informative features and to suppress less useful ones was introduced into the bottleneck. To avoid gradient disappearance, between the input of the SE–ResNet Bottleneck module and its output, a 1 × 1 convolution layer was used to adjust the number of channels if the module was at the top of its stage; otherwise, an identity shortcut link was used to allow gradient propagation [23]. Following the last SE–ResNet Bottleneck module, the image was fed to a global pooling layer and a dropout layer, which helped to avoid overfitting. The final output layer (fully connected layer) was activated by using the softmax function, which provided a probability of PSVT. The architecture of the model is shown in Figure 3.

### 2.4. Outcomes of Interest

In addition, we created ROC curves and measured the corresponding AUCs for the validation and testing sets to assess the classification performance of the proposed CNN to screen patients for PSVT based on ECG images during normal sinus rhythm. We selected a suitable probability threshold on the ROC curve for the validation set and applied the same threshold to the testing set to calculate the F1 score, accuracy, sensitivity, specificity and precision. Moreover, we created receiver–operator curves and measured the corresponding AUCs to assess the network strength of the proposed CNN model to screen patients for PSVT based on ECG data alone.

### 2.5. Statistical Analysis

Descriptive statistics were applied to report the clinical characteristics of patients included in this study. Continuous variables were expressed as mean values ± standard deviation. Categorical variables were expressed as ratios or percentages. Levene’s test was used to check the homogeneity of variance. Normally distributed data were compared using independent student’s *t*-test. Chi-square was used for categorical variables. Statistical optimization of the CNN was performed through iterative training using the Keras package. Measures of diagnostic performance included the ROC AUC, accuracy, sensitivity, specificity and the F1 score. We used two-sided 95% CIs to summarize the sample variability in the estimates. SPSS version 19.0 (Armonk, NY, USA: IBM Corp) was used for statistical analysis. All tests were performed with a two-tailed significance level of 0.05.

## 3. Results

### 3.1. Dataset Characteristics

We screened 407 ECG images in the PSVT group and 1794 ECG images in the control group according to the inclusion and exclusion criteria. The mean age of patients was 48.2 ± 16.4 years, and 46.3% of patients were male. There were no statistically significant differences in ECG characteristics between the control group and the PSVT group in terms of P–R interval, QRS interval or QT interval. However, the heart rates and QTc were slightly higher in the control group than in the PSVT group. The clinical characteristics of these patients are shown in Table 1.

### 3.2. Model Screening Performance

Receiver–operator curves (ROCs) were created, and the areas under the curves (AUCs) were measured to assess the network. The AUC for detecting PSVT was 0.956 (0.917–0.996) when using the validation set and 0.975 (0.959–0.991) when using the testing set (Figure 4). The probability value that yielded preferable sensitivity, specificity and accuracy of 95.5% on the validation set was applied to the testing set and yielded an F1 score of 88.0%, sensitivity of 90.2% (81.2–95.4), specificity of 96.6% (94.0–98.2), precision of 86.0% (76.5–92.3) and an overall accuracy of 95.5%. The proposed CNN model provided a low-cost, non-invasive and feasible method to diagnose PSVT in patients with palpitations.

## 4. Discussion

In the past 10 years, several studies have been conducted attempting to detect arrhythmias and cardiac arrest based on CNN-enabled ECG, and have shown promising performance. In the present study, we trained a CNN model that automatically extracts the hidden features from ECG images to diagnose unrecognized PSVT based on sinus-rhythm ECGs. This model demonstrated an overall accuracy of 95.5%, sensitivity of 90.2%, and specificity of 96.6% to predict pre-onset PSVT from ECGs. Our results provide an effective and convenient method for the early detection of PSVT from palpitation patients, which could help these patients receive timely intervention and improve their quality of life.

PSVT is a clinical syndrome characterized by the presence of a regular and rapid heart rhythm with abrupt onset and termination [2]. By virtue of its episodic nature, confirming a diagnosis of PSVT can be challenging [4]. Patients may be initially misdiagnosed with anxiety or other rhythm disorders. Consequently, PSVT can go undiagnosed for years, and patients may be diagnosed with other cardiac conditions or arrhythmias before obtaining a formal diagnosis of PSVT [2]. Furthermore, some patients need invasive electrophysiological study to confirm PSVT. Developing low-cost and non-invasive methods to detect PSVT has important diagnostic and therapeutic implications.

ECG is a common, noninvasive examination method to detect cardiovascular diseases. With the rapid advances in artificial intelligence (AI) in ECG interpretation, AI cannot only make classification based on ECGs, but it can also detect the subtle signals in the ECG that might be invisible to the human eye, yet which contain important information to predict diseases [13,24]. An AI-enabled ECG using a convolutional neural network could detect the electrocardiographic signature of atrial fibrillation present during normal sinus rhythm using standard 10-s, 12-lead ECGs with an AUC of 0.87, sensitivity of 79.0%, specificity of 79.5%, F1 score of 39.2% and overall accuracy of 79.4% [20]. Jo et al. trained a deep learning model (DLM) that could identify PSVT during normal sinus rhythm. During accuracy testing, the ROC of the DLM was 0.966, the accuracy, sensitivity, specificity, positive predictive value and negative predictive value of the DLM were 0.970, 0.868, 0.972, 0.255 and 0.998, respectively. They also found that the QT interval is highly correlated with the development of PSVT using sensitivity map [25]. In the present study, we trained a CNN with sinus-rhythmic ECGs of PSVT patients and control patients with no signs of PSVT. This CNN algorithm showed an AUC of 0.975, overall accuracy of 95.5%, sensitivity of 90.2%, specificity of 96.6%, precision of 86.0% and F1 score of 88.0%, which was consistent with the DLM. Our data also support the idea that QT interval is highly correlated with the development of PSVT, as there was a statistical difference of QTc between the control group and the PSVT group. Our study is also different from Jo’s study. Firstly, only AVNRT and concealed accessory-pathway-induced AVRT, which could not be diagnosed with sinus-rhythmic ECGs, were included, and overt accessory-pathway-induced AVRT (WPW syndrome), which could be diagnosed with sinus-rhythmic ECGs by a cardiologist, was excluded in our study. Secondly, in the present study, the cases in the PSVT group were confirmed by electrophysiological study and radiofrequency ablation to exclude atrial tachycardia. However, both of these models could benefit clinicians in the future application of sinus-rhythmic ECGs to diagnose unrecognized PSVT in a timely manner, and would also avoid unnecessary invasive electrophysiology studies, which would ease the pain of patients and reduce medical expenditure.

In previous studies, raw data of ECGs were utilized for artificial intelligence-based ECGs to identify cardiovascular diseases [19,20,25]. Our group developed an AI-based method to screen patients with left ventricular ejection fraction (LVEF) of 50% or less using ECG data alone. The CNN algorithm showed an overall accuracy of 73.9%, sensitivity of 69.2%, specificity of 70.5%, positive predictive value of 70.1% and precision of 69.9%, which demonstrates that a well-trained CNN algorithm may be used as a low-cost and noninvasive method to identify patients with left ventricular dysfunction [18]. In the present study, we used images of standard 10-s, 12-lead ECGs for input of training, validation and testing. Our data demonstrated that ECG images and not only raw ECG data from the ECG management system could be applied for deep learning training to predict arrhythmias with sinus-rhythm ECGs. Contrary to ECG raw data, ECG images are much more easily acquired for patients examined at different hospitals with different ECG machines. Once a CNN network is trained, it can be applied to any standard 12-lead ECG images from a mobile device. In the future, patients with suspected PSVT could use this low-cost and noninvasive tool to exclude or confirm pre-onset PSVT, which is especially beneficial for patients from rural areas or developing countries.

### Limitations

This study has several limitations. Firstly, we cannot know exactly what features are extracted by the algorithm, which is known as a “black box” system. Secondly, the number of ECGs for PSVT patients was relatively small, which might limit the model performance and reduce model robustness. Finally, this is a single-center retrospective study. Prospective, large-scale and multi-center studies are required to validate the performance of the model.

## 5. Conclusions

An artificial-intelligence-enabled ECG acquired during normal sinus rhythm can identify individuals with a high likelihood of PSVT. The results of this study could have useful implications for PSVT screening and diagnosis.

## Figures and Tables

**Figure 1 jcm-11-04578-f001:**
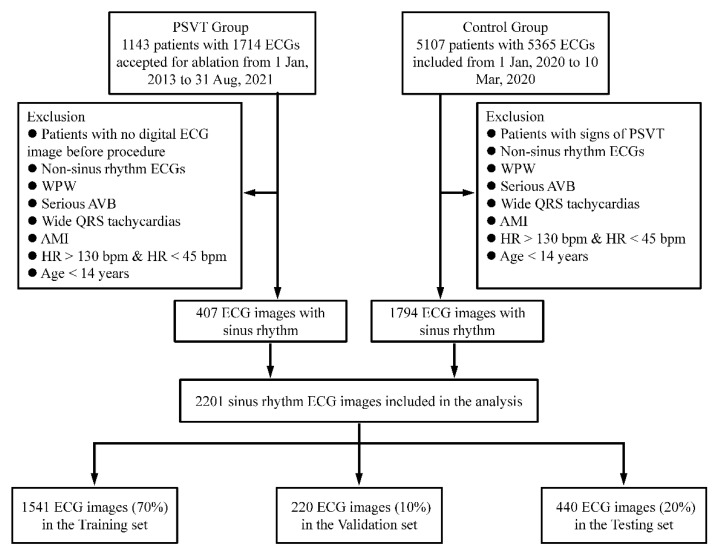
Flowchart of data collection and dataset creation. A total of 1714 ECGs of 1143 patients with PSVT and 5365 ECGs of 5107 control patients were collected. After exclusion of non-sinus-rhythm ECGs, WPW syndrome, serious AVB, wide QRS tachycardias, acute myocardial infarction, HR > 130 bpm, HR < 45 bpm and age < 14 years, a total of 2201 ECGs (407 ECGs for PSVT group and 1794 ECGs for control group) were included in this study. These ECGs were randomly divided into three datasets: a training set (n = 1541), validation set (n = 220) and testing set (n = 440).

**Figure 2 jcm-11-04578-f002:**
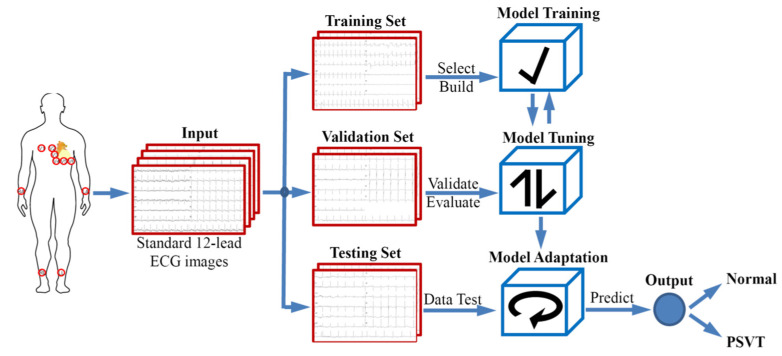
The comprehensive process of the creation and evaluation of the CNN model. The ECG images were acquired and allocated to the training set, validation set, or testing set. ECG images in the training set were used as the input of the CNN, whereas the testing set was used to evaluate the screening performance of the CNN model.

**Figure 3 jcm-11-04578-f003:**
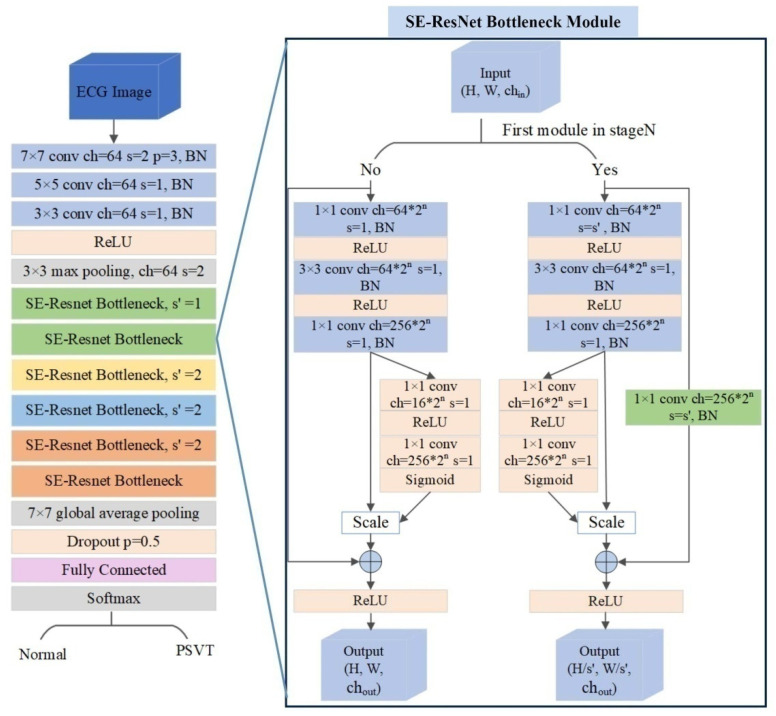
The architecture of the artificial intelligence model. The SE–ResNet Bottleneck contained 4 stages (State N: Stage 0, green; Stage 1, yellow; Stage 2, blue; and Stage 3, orange). Stage 0 and Stage 3 had two SE–ResNet Bottleneck modules each, and Stage 1 and Stage 2 had only one SE–ResNet Bottleneck module each. The value of N ranged from 0 to 3, which also determined the number of channels per module.

**Figure 4 jcm-11-04578-f004:**
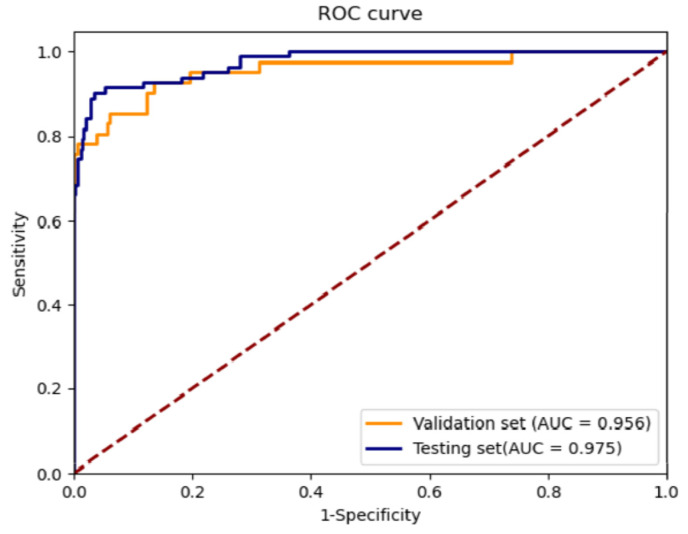
ROC of the screening performance on the validation set and testing set: AUC, area under the curve; ROC, receiver–operating characteristic curve.

**Table 1 jcm-11-04578-t001:** Baseline characteristics of patients and ECGs.

Parameters	Control (n = 1794)	PSVT (n = 407)	Total (n = 2201)	*p*-Value
Age (years)	47.2 ± 16.5	52.7 ± 14.9	48.2 ± 16.4	<0.001
Gender (male, %)	835 (46.5)	183 (45.0)	1018 (46.3)	0.564
Heart rate	80.6 ± 14.7	78.0 ± 18.8	80.1 ± 14.6	0.001
P–R interval	149.5 ± 19.1	151.0 ± 22.3	149.8 ± 19.7	0.189
QRS interval	84.7 ± 11.7	85.1 ± 11.7	84.8 ± 11.7	0.466
QT interval	363.6 ± 32.0	361.0 ± 33.1	363.1 ± 32.2	0.134
QTc	416.7 ± 25.9	411.3 ± 26.7	415.8 ± 26.1	<0.001

## Data Availability

The data that support the findings of this study are available from the corresponding author upon reasonable request.

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
