# Peer review of "Deep-Learning-Based Detection of Paroxysmal Supraventricular Tachycardia Using Sinus-Rhythm Electrocardiograms"

_jcm, 2022, doi:10.3390/jcm11154578_

Round 1
Reviewer 1 Report
In this study, the authors propose to develop a deep learning model for prediction of PSVT. The authors develop and validate the proposed model on an ECG dataset collected at The Affiliated Wuxi People's Hospital of Nanjing Medical University. My general comments are as follows:
1) The manuscript requires extensive editing to improve its grammar. There are many grammatical mistakes throughout the paper.
e.g. Line 59, "relys" should be "relies".
2) The authors have only included results of their proposed model. I would recommend the authors to include a few more baseline model comparison. This is to ensure that the proposed CNN model is indeed the best choice of model for the prediction task
3) It is my understanding that the authors are performing a "detection" of PSVT and not "prediction" of PSVT. In a prediction setting, the condition is not present yet, and a prediction horizon is usually defined. No such horizon is defined in the paper. I suggest modifying the paper to reflect the fact it is a detection of PSVT paper and not prediction of PSVT
4) Can the authors comment on how their paper is different (in terms of study hypothesis, study design etc.) from a similarly published paper - Artificial intelligence to diagnose paroxysmal supraventricular tachycardia using electrocardiography during normal sinus rhythm? (https://academic.oup.com/ehjdh/article/2/2/290/6131636)
Author Response
A point-by point response to reviewer’s comments
In this study, the authors propose to develop a deep learning model for prediction of PSVT. The authors develop and validate the proposed model on an ECG dataset collected at The Affiliated Wuxi People's Hospital of Nanjing Medical University. My general comments are as follows.
Response: We appreciate the reviewer for your constructive comments.
Comment 1: The manuscript requires extensive editing to improve its grammar. There are many grammatical mistakes throughout the paper.
e.g. Line 59, "relys" should be "relies".
Response: Thanks for your suggestion. We checked our manuscript carefully and revised three grammatical mistakes: "relys" to "relies" (Page 2), "payed" to "paid" (Page 4) and "summarise" to "summarize" (Page 5). We also consulted Language Editing Services via MDPI and revised our manuscript according to their editing.
Comment 2: The authors have only included results of their proposed model. I would recommend the authors to include a few more baseline model comparison. This is to ensure that the proposed CNN model is indeed the best choice of model for the prediction task.
Response: We thank the reviewer for this constructive comment. As a update of our AI station, the old system ‘Windows Server 2012 R2 with NVIDIA Tesla V100 (16 GB)’ was changed into ‘Linux centos 7.8 with NVIDIA PCIe A100 (40GB)’. We trained both baseline model SE-ResNet50 and the proposed CNN model on the new AI station. Considering the ECG image was consisted of multiple waves, we paid more attention to the edge detection and we set two more convolution layers in the initial stage of the network to extract waveform features in the proposed CNN model. In addition, to reduce over-fitting, we reduced 10 SE-ResNet bottleneck modules. The network with parameters corresponding to the best outcome on validation dataset was selected. The best outcome indicated the maximal sensitivity with the cut-off threshold of specificity above 97.0% for each model. Then, the network with parameters was performed on the testing dataset. As shown in Figure A and B, the AUC for detecting PSVT were 0.961 with SE-ResNet50 and 0.981 with our proposed CNN model when using the testing set. The F1 score, accuracy, sensitivity, specificity and precision were shown in the table. Our proposed CNN model demonstrated a slight better performance on PSVT detection task but it spent shorter time.
Figure. ROC of SE-ResNet50 and the proposed CNN model performance on the validation set and testing set.
Table: Comparison of SE-ResNet50 and the proposed CNN model
|
Models |
F1 score |
Accuracy |
Sensitivity |
Specificity |
Precision |
|
SE-ResNet50 |
0.871 |
0.952 |
0.866 |
0.972 |
0.877 |
|
Proposed CNN model |
0.898 |
0.961 |
0.915 |
0.972 |
0.882 |
Comment 3: It is my understanding that the authors are performing a "detection" of PSVT and not "prediction" of PSVT. In a prediction setting, the condition is not present yet, and a prediction horizon is usually defined. No such horizon is defined in the paper. I suggest modifying the paper to reflect the fact it is a detection of PSVT paper and not prediction of PSVT.
Response: We agree with your opinion and thanks for your suggestion. We revised the whole manuscript and changed prediction into detection. And we also revised the title as ‘Deep learning-based detection of paroxysmal supraventricular tachycardia using sinus rhythm electrocardiograms’.
Comment 4: Can the authors comment on how their paper is different (in terms of study hypothesis, study design etc.) from a similarly published paper - Artificial intelligence to diagnose paroxysmal supraventricular tachycardia using electrocardiography during normal sinus rhythm? (https://academic.oup.com/ehjdh/article/2/2/290/6131636)
Response: We thank the reviewer for this insightful comment and providing us the website as we did not find this paper on Pubmed. We find this paper and read it carefully. In the paper, the authors trained a deep learning model to diagnose paroxysmal supraventricular tachycardia using electrocardiography during normal sinus rhythm. The proposed deep learning model demonstrated a high performance in identifying PSVT during normal sinus rhythm. It was different from our study. First, the term of “PSVT” in the present study refers to atrioventricular nodal re-entry tachycardia (AVNRT) or concealed accessory pathway induced atrioventricular re-entry tachycardia (AVRT). Overt accessory pathway (Wolff–Parkinson–White syndrome) was included in that study, while it was excluded in our study as it could be diagnosed with sinus rhythmic ECG by cardiologist. Second, in the present study, the cases in the PSVT group were confirmed by electrophysiological study and radiofrequency ablation, only AVNRT and concealed accessory pathway that could not be diagnosed with sinus rhythmic ECG were induced. In Jo’s paper, patients with ECG showing PSVT were defined as ‘positive ECG’ (PSVT group). As we know, ECG showing PSVT could not rule out atrial tachycardia. Third, we used ECG images for model training while they used ECG raw data. Our data demonstrated that ECG images but not only raw ECG data from ECG management system could be applied for deep learning training to detect arrhythmias with sinus rhythmic ECGs. Contrasts to ECG raw data, ECG images are much more easily acquired for patients examined from different hospitals with different ECG machine. Forth, they performed a multi-center study with external validation, while our study was a single-center study. However, our results were consistent with this study. And they also found that QT interval was highly correlated with the development of PSVT using sensitivity map. Our data supported this idea as there was a statistical difference between control group and PSVT group.
We thank the reviewers for your great efforts in improving our manuscript and we hope that our revisions have fully addressed all concerns.

Reviewer 2 Report
I had the pleasure of reviewing this study by Wang and colleagues that endeavours to use CNN to categorise 12-lead ECGs of individuals who have previously suffered an episode of PSVT compared with ECGs of control individuals who have not. This study comprehensively describes its methodology, a feature that is often lacing with similar manuscripts. However, I think that there are more limitations to the study than those that the authors describe and as such I would suggest tempering the author’s conclusions accordingly. Most of the manuscript is well written, but occasionally there are grammatical and style errors that hamper the meaning of the manuscript.
I have made the following comments:
Abstract
- It is not clear in the abstract what the PSVT patients are. I suggest being more clear that PSVT has been detected, they have returned to sinus and they are awaiting an electrophysiological procedure.
- Perhaps “decline of quality of life” should be ‘decline in quality of life’
- Line 62, “automatically even subtle features” should have a comma after automatically because there is a natural break in the sentence.
Introduction
- Probably don’t need to refer to what is effectively inclusion criteria in the introduction
- Hypothesis sentence does not make sense. I would recommend re-writing.
- The following sentence is probably more appropriate in the results than in the hypothesis/ aims section of the introduction. “The proposed CNN model provided a low-cost, non-invasive and feasibility method for palpitation patients with latent PSVT patients to diagnose PSVT”. I suggest this is moved to the results.
Methods
- Throughout the paper I think the authors need to be more clear about the ‘PSVT patient’ ECGs. It is not immediately clear at what point the ECGs are taken.
- “were included the study they had a sinus rhythm ECG before the procedure.” I presume should be ‘were included the study if they had a sinus rhythm ECG before the procedure.’
- Perhaps more detail on where the control group are from? Were they suspected to have PSVT? It is important to understand whether the control ECGs are valid controls.
- Perhaps clearly separate your outcomes of interest by a subheading from the description of the AI model? I suggest adding a subheading such as Outcomes of interest above line 177.
Results
- Line 216 “precision of 86•0%” perhaps should be ‘precision of 86.0%’
Discussion
- How could the model be improved? Further validated?
- I am surprised that the authors have not referred to a similar study by Yong-Yeon Jo and colleagues published recently in the EHJ Digital Health who also look to predict PSVT using 12-lead ECGs in sinus rhythm. I would suggest drawing parallels and comparing findings
Limitations
- Control from different years. There could be differences such as ECG quality.
- The fact that baseline characteristics are significantly different between control for several variables, particularly that of age
- A potential limitation is that whilst the PSVT group were in sinus rhythm before the procedure, presumably by this point the PSVT had already been detected clinically. Therefore, prediction at this stage would not be useful to further patients and the ECG may have more subtle pathological signs that it may have had before PSVT was detected clinically when the patient was more healthy. It would be useful if this model was applied to patients before the PSVT is detected to determine whether the model can be used to detect PSVT before it is picked up clinically.
Conclusion
- The authors have quite rightly pointed out that this study is relatively small and retrospective, as such I would suggest toning down their conclusions such that the study is referred to as a ‘pilot’ study. This seems in keeping with the overall aims that the authors describe.
Author Response
A point-by point response to reviewer’s comments
I had the pleasure of reviewing this study by Wang and colleagues that endeavours to use CNN to categorise 12-leadï¼› ECGs of individuals who have previously suffered an episode of PSVT compared with ECGs of control individuals who have not. This study comprehensively describes its methodology, a feature that is often lacing with similar manuscripts. However, I think that there are more limitations to the study than those that the authors describe and as such I would suggest tempering the author’s conclusions accordingly. Most of the manuscript is well written, but occasionally there are grammatical and style errors that hamper the meaning of the manuscript. I have made the following comments:
Response: We appreciate the reviewer for your critical and constructive comments.
Abstract
Comment 1: It is not clear in the abstract what the PSVT patients are. I suggest being more clear that PSVT has been detected, they have returned to sinus and they are awaiting an electrophysiological procedure.
Response: We thank the reviewer for your suggestion. We revised the Methods in the Abstract as ‘In the current study, convolution neural network (CNN) was trained with normal sinus rhythmic, standard 12-lead electrocardiographs (ECGs) of negative control patients and PSVT patients to identify patients with latent PSVT. PSVT was referred to atrioventricular nodal re-entry tachycardia and atrioventricular re-entry tachycardia based on concealed accessory pathway, confirmed by electrophysiological procedure. Negative control group data was obtained from 1794 revisited patients with at least one normal sinus rhythmic ECG without any palpitation symptom.’
Comment 2: Perhaps “decline of quality of life” should be ‘decline in quality of life’.
Response: We revised ‘decline of quality of life’ into ‘decline in quality of life’.
Comment 3: Line 62, “automatically even subtle features” should have a comma after automatically because there is a natural break in the sentence.
Response: We add a comma after automatically.
Introduction
Comment 1: Probably don’t need to refer to what is effectively inclusion criteria in the introduction
Response: Thank you for your suggestion. We deleted ‘from The Affiliated Wuxi People's Hospital of Nanjing Medical University’ in the Introduction and revised as ‘To test and verify our hypothesis, we trained, validated and tested a CNN model using patients with standard 12-lead, normal sinus rhythmic ECGs’.
Comment 2: Hypothesis sentence does not make sense. I would recommend re-writing.
Response: We thank the reviewer for your suggestion. We revised the hypothesis as ‘we hypothesized that a deep neural network could identify PSVT patients even they were not in the onset stage.’
Comment 3: The following sentence is probably more appropriate in the results than in the hypothesis/ aims section of the introduction. “The proposed CNN model provided a low-cost, non-invasive and feasibility method for palpitation patients with latent PSVT patients to diagnose PSVT”. I suggest this is moved to the results.
Response: We thank the reviewer for this thoughtful suggestion. We moved the sentence to the results.
Methods
Comment 1: Throughout the paper I think the authors need to be more clear about the ‘PSVT patient’ ECGs. It is not immediately clear at what point the ECGs are taken.
Response: We thank the reviewer for this important comment. The PSVT group inclusion criteria were as follows: (1) Patient were included if they had abrupt onset and termination of palpitation symptoms or they were diagnosed as PSVT clinically; (2) Patients were diagnosed and confirmed with PSVT by electrophysiological study and radiofrequency ablation; (3) Patient were included if they had a sinus rhythmic ECG before the electrophysiological procedure.
Comment 2: “were included the study they had a sinus rhythm ECG before the procedure.” I presume should be ‘were included the study if they had a sinus rhythm ECG before the procedure.’
Response: We agree with your opinion and we revise this sentence as ‘were included in the study if they had a sinus rhythmic ECG before the electrophysiological procedure’.
Comment 3: Perhaps more detail on where the control group are from? Were they suspected to have PSVT? It is important to understand whether the control ECGs are valid controls.
Response: We appreciate the reviewer for this constructive comment. As described in Methods: 5365 ECGs of 5107 patients were also collected between January 1, 2020 and March 10, 2020 as control group. For the control group, patients that were evaluated without evidence of PSVT in the outpatient clinic by cardiologist via history collection, medical records or telephone follow-up were included.
Comment 4: Perhaps clearly separate your outcomes of interest by a subheading from the description of the AI model? I suggest adding a subheading such as Outcomes of interest above line 177.
Response: We thank the reviewer for your suggestion. We add ‘Outcomes of interest’ as a subheading in your indicated location.
Results
Comment 1: Line 216 “precision of 86•0%” perhaps should be ‘precision of 86.0%’.
Response: We are sorry for this mistake. We revised ‘precision of 86•0%’ as ‘precision of 86.0%’.
Discussion
Comment 1: How could the model be improved? Further validated?
Response: We thank the reviewer for this constructive comment. We improved our model on the basis of SE-ResNet50. Considering the ECG image was consisted of multiple waves, we paid more attention to the edge detection and we set two more convolution layers in the initial stage of the network to extract waveform features in the proposed CNN model. In addtion, to reduce over-fitting, we reduced 10 SE-ResNet bottleneck modules.
As a update of our AI station, the old system ‘Windows Server 2012 R2 with NVIDIA Tesla V100 (16 GB)’ was changed into ‘Linux centos 7.8 with NVIDIA PCIe A100 (40GB)’. We trained both baseline model SE-ResNet50 and the proposed CNN model on the new AI station. The network with parameters corresponding to the best outcome on validation dataset was selected. The best outcome indicated the maximal sensitivity with the cut-off threshold of specificity above 97.0% for each model. Then, the network with parameters was performed on the testing dataset. As shown in Figure A and B, the AUC for detecting PSVT were 0.961 with SE-ResNet50 and 0.981 with our proposed CNN model when using the testing set. The F1 score, accuracy, sensitivity, specificity and precision were shown in the Table. Our proposed CNN model demonstrated a slight better performance on PSVT detection task but it spent shorter time.
Figure. ROC of SE-ResNet50 and the proposed CNN model performance on the validation set and testing set.
Table: Comparison of SE-ResNet50 and the proposed CNN model
|
Models |
F1 score |
Accuracy |
Sensitivity |
Specificity |
Precision |
|
SE-ResNet50 |
0.871 |
0.952 |
0.866 |
0.972 |
0.877 |
|
Proposed CNN model |
0.898 |
0.961 |
0.915 |
0.972 |
0.882 |
Comment 2: I am surprised that the authors have not referred to a similar study by Yong-Yeon Jo and colleagues published recently in the EHJ Digital Health who also look to predict PSVT using 12-lead ECGs in sinus rhythm. I would suggest drawing parallels and comparing findings.
Response: We thank the reviewer for finding this paper for us. We started this study since Jan 2020 and we did not find this paper in Pubmed when we prepared this manuscript one year ago. Now we read this paper carefully and find it is a similar study with our manuscript that we should site.
In the paper, the authors trained a deep learning model to diagnose paroxysmal supraventricular tachycardia using electrocardiography during normal sinus rhythm. The proposed deep learning model demonstrated a high performance in identifying PSVT during normal sinus rhythm. Our study was consistent with this study. And they also found that QT interval was highly correlated with the development of PSVT using sensitivity map. Our data supported this idea as there was a statistical difference between control group and PSVT group.
However, our study is also different from this paper. First, the term of “PSVT” in the present study is referred to atrioventricular nodal re-entry tachycardia (AVNRT) or concealed accessory pathway induced atrioventricular re-entry tachycardia (AVRT). Overt accessory pathway (WPW syndrome) was included in that study; while it was excluded in our study as it could be diagnosed with sinus rhythm ECG by cardiologist. Second, in the present study, the cases in the PSVT group were confirmed by electrophysiological study and radiofrequency ablation, only AVNRT and concealed accessory pathway that could not be diagnosed with sinus rhythmic ECG were induced. In Jo’s paper, patients with ECG showing PSVT were defined as ‘positive ECG’ (PSVT group). As we know, ECG showing PSVT could not exclude atrial tachycardia. Third, we used ECG images for model training while they used ECG raw data. Our data demonstrated that ECG images but not only raw ECG data from ECG management system could be applied for deep learning training to predict arrhythmias with sinus rhythmic ECGs. Contrasts to ECG raw data, ECG images are much more easily acquired for patients examined from different hospitals with different ECG machine. Forth, they performed a multi-center study with external validation, while our study was a single-center study.
We cited this paper (Ref 25) and revised discussion section accordingly.
Limitations
Comment 1: Control from different years. There could be differences such as ECG quality.
Response: We appreciate the reviewer for this constructive comment. Indeed, ECGs of Control group from different years may have a bias on the outcome and affect the performance of the model. However, ECG data was collected with same ECG machine although they were from different years. What’s more, All ECG images were used for input of deep learning model with same size and resolution so as to reduce the influence of ECG quality on the performance of the proposed CNN model.
Comment 2: The facts that baseline characteristics are significantly different between control for several variables, particularly that of age.
Response: We thank the reviewer for this important comment. Indeed, there were significantly differences of several baseline variables between control and PSVT group, which might influence outcome of detection. However, it was difficult to avoid this bias as it was real and unselected data. Bigger qualified dataset was needed to solve this question. In addition, QTc that was significantly different between two groups was highly correlated with the development of PSVT using sensitivity map reported by Jo et al, which might be the key detection characteristics of deep learning model in our study.
Comment 3: A potential limitation is that whilst the PSVT group were in sinus rhythm before the procedure, presumably by this point the PSVT had already been detected clinically. Therefore, prediction at this stage would not be useful to further patients and the ECG may have more subtle pathological signs that it may have had before PSVT was detected clinically when the patient was more healthy. It would be useful if this model was applied to patients before the PSVT is detected to determine whether the model can be used to detect PSVT before it is picked up clinically.
Response: We thank the reviewer for this thoughtful comment. Yes, you are right. It would be useful if this model could be used to detect PSVT before it is picked up clinically, which was the aim of our study. All the PSVT patients that were included in our study were diagnosed and confirmed by electrophysiological study and radiofrequency ablation. Partial of the PSVT patients with palpitation were not documented with PSVT ECGs, which meant these patients were not diagnosed clinically before the procedure. The sinus rhythmic ECGs of these patients before the procedure were used to train deep learning model. Therefore, based on the outcome of our study, the proposed deep learning model could identify the patients with already suffered PSVT but was not detected clinically before.
Conclusion
Comment 1: The authors have quite rightly pointed out that this study is relatively small and retrospective, as such I would suggest toning down their conclusions such that the study is referred to as a ‘pilot’ study. This seems in keeping with the overall aims that the authors describe.
Response: We appreciate the reviewer for your constructive comments. We revised the conclusion of Abstract as: Our study reveals that a well-trained CNN algorithm may be a rapid, effective, inexpensive and reliable method which would contribute to an early detection of PSVT. We also revised the conclusion of Manuscript as: In conclusion, an artificial intelligence-enabled ECG acquired during normal sinus rhythm permits identification of individuals with a high likelihood of PSVT. This result could have useful implications for PSVT screening and diagnosis.
We thank the reviewers for your great efforts in improving our manuscript and we hope that our revisions have fully addressed all concerns.